# Method to Generate Chlorine Dioxide Gas In Situ for Sterilization of Automated Incubators

**DOI:** 10.3390/pathogens13111024

**Published:** 2024-11-20

**Authors:** Cédric Schicklin, Georg Rauter, Philippe Claude Cattin, Manuela Eugster, Olivier Braissant

**Affiliations:** 1Bio-Inspired RObots for MEDicine-Laboratory, Department of Biomedical Engineering, University of Basel, 4123 Allschwil, Switzerland; cedric.schicklin@unibas.ch (C.S.); 2Center for medical Image Analysis & Navigation, Department of Biomedical Engineering, University of Basel, 4123 Allschwil, Switzerland; philippe.cattin@unibas.ch; 3Neuro Robotics Group, ARTORG Center, University of Bern, 3008 Bern, Switzerland; 4Department of Neurosurgery, Inselspital, Bern University Hospital, 3008 Bern, Switzerland; 5Biological Calorimetry Lab, Department of Biomedical Engineering, University of Basel, 4123 Allschwil, Switzerland

**Keywords:** decontamination, chlorine dioxide, bacteria, fungi, viruses

## Abstract

Pharmaceutical preclinical tests using cell cultures are nowadays commonly automated. Incubator microbial contaminations impact such tests. Chlorine dioxide (ClO_2_) is widely used in aqueous solutions. However, a gaseous form, such as chlorine dioxide gas (gClO_2_), can effectively access unreachable spaces, such as closed cell culture incubators. Steam sterilization requires a temperature rise to at least 121 °C, thus limiting the possibility of automation elements for sensors and actuators. gClO_2_ sterilization is an ambient-temperature sterilization method. This article aims to demonstrate that gClO_2_ generated from solid powder tablets is efficient for sterilizing incubators and can be automated. We selected (i) *Bacillus subtilis* strain, (ii) *Saccharomyces cerevisiae*, and (iii) T7 phages as representatives for (i) bacteria, (ii) fungi, and (iii) viruses for each domain to evaluate the sterilization efficiency. This study demonstrated that gClO_2_ can be generated inside the incubator from a solid powder tablet without specific equipment and can effectively fight biological proxies in 15 min. After 30 sterilization cycles, the actuators and sensors mounted inside the incubator were still operating. Our proposed sterilization method seems to be generally applicable for automated in situ sterilization of incubators and medical robots.

## 1. Introduction

Cell cultures, organoids, and Organs-on-a-Chip have become essential to ensure pharmaceutical preclinical safety tests. They are grown inside incubators that maintain optimal temperature, humidity, and carbon dioxide content of the inner atmosphere. However, pharmaceutical preclinical safety tests are often work-intensive and require attention for a medium change, growth monitoring, or addition of growth factors. Therefore, many attempts are made to automate such testing [1]. Automated incubators are expected to increase test throughput and data quality in this context. Indeed, automated systems integrate sensors, control systems, and actuators designed to perform a function with minimal or no human intervention, thus decreasing the required workforce and the risk of human error or contamination. With the rising complexity of organ models used for pharmaceutical testing over time, automated incubator systems were equipped with perfusion sub-systems, mainly for perfused Organ-on-a-Chip models, such as endothelium models. The perfusion sub-system is technically realized using a tilting station or a perfusion pump. In an automated incubator, the consequences of microbial contaminations can severely impact in vitro experiments (safety test, Organ-on-a-Chip, and basic research) [2] and lead to not only a loss of data but also money and time, as the models mentioned above are of long duration (in order of weeks) [3,4] and sometimes difficult to grow until mature for pharmaceutical tests. Microbial decontamination of automated incubators between different in vitro assays is thus critical to ensure the reproducibility of experiments.

Similarly, in the future, such decontamination will be required to avoid contamination of engineered tissues or organs intended for patient transplants. In this context, microbial decontamination is crucial to ensure the safe operation of automated incubators. Sterilization automation contributes to the increase in automated incubator test throughput, as minimal or no human intervention is required. In this way, sterilization protocols might be performed automatically or semi-automatically and rapidly between measurements, overnight, or during non-operation time.

Laboratory personnel commonly use aqueous-based disinfectants to perform sterilization. Alcohol, mixed with water, is widely used as a disinfectant in pharmaceutical laboratories. Alternatively, aqueous chlorine-based disinfectants are also used [5,6]. Sodium hypochlorite, commonly known as bleach, is a widely used household product and is an example of aqueous chlorine-based disinfectant.. A chlorine-based disinfectant is also used for the treatment of drinking water [7] and is demonstrated to have antimicrobial activity against bacteria [7,8,9], fungi [10], and viruses [11,12]. Aqueous-based disinfectants have a high efficacy, theoretically [13], but they require access to every part of the system to apply the disinfectant with a wipe or spray [14], as well as access for removing excess aqueous-based disinfectant fluid. However, access to every system part is rarely available for automated incubators. Also, aqueous disinfectants have difficulties reaching cramped spaces, such as protected electrical circuits or inside electrical connectors. Aqueous sterilization is also time-consuming for laboratory and hospital personnel.

Vapor- or gas-based sterilization is commonly used to overcome the accessibility issue of applying aqueous-based disinfectants [15]. Steam-based sterilization is a popular gas-based method to sterilize laboratory instruments. Steam-based sterilization of incubators requires increasing the incubator air temperature to at least 121 °C for 3 min [14]. The high temperature and pressure used in steam sterilization might damage many components of automated systems and thus limit the selection of such components (in particular, sensors or actuators) for automated systems. Furthermore, steam sterilization is time-consuming, as it can take several hours to achieve the desired sterilization temperature and cool down to the test temperature. Gaseous chemical disinfectants might be used for sterilization as an alternative for instruments incompatible with steam-based sterilization temperatures. Formaldehyde and ethylene oxide are effective but flammable and highly toxic. Hydrogen peroxide vapor decomposes into water and oxygen, leaving no toxic residues, and does not require ventilation. Hydrogen peroxide vapor is more commonly used as an ambient-temperature chemical disinfectant [16], but it requires an external fumigation device, called a fumigator, to generate the vapor. One hundred minutes are required to disinfect a room-size volume with hydrogen peroxide [17]. Using a fumigator to generate the disinfectant vapors opens new challenges to transferring the disinfectant vapors from the fumigator to the incubator. Gaseous ClO_2_ (gClO_2_) was demonstrated to have antimicrobial effects [15,16,18,19,20], but most of the time, an external fumigation system is required [21,22], and several hours are necessary for the fumigation of an entire room [18].

While gClO_2_ has desired antimicrobial effects, respiratory tract toxicity occurs at low concentrations, such as 0.5 PPM [23,24]. The above sterilization methods include safety risks for the laboratory personnel, such as potential contact with hot parts or exposure to toxic gases. Therefore, automation of sterilization is also beneficial in reducing risks for laboratory personnel and improving the general environment’s health and safety.

This publication investigates a method to generate gClO_2_ for sterilization from a solid powder tablet and its applicability for use in situ within an automated perfusion incubator without external devices. This approach eliminates the need to use an external fumigator or additional system while ensuring laboratory personnel safety, sterilization efficacy, and fast turnover between pharmaceutical preclinical safety tests.

## 2. Materials and Methods

### 2.1. Biological Proxies

The goal was to demonstrate that gClO_2_, combined with the proposed sterilization protocol, is effective against most known biological indicators that mimic contamination. As the method has not been tested extensively, the biological indicator used for the experiments was biosafety level 1 (BS1). Therefore, the sterilization’s effectiveness was qualified without dangerous microbes. The efficacy of gClO_2_ was tested using conventional biological indicators from three domains: (i) bacteria, (ii) fungi, and (iii) viruses. *Bacillus subtilis* was selected for the prokaryote group. *Saccharomyces cerevisiae* was chosen for the group of eukaryotes, and the bacteriophage T7 was chosen for the virus group using *Escherichia coli* as a host. While testing this sterilization protocol against all known biological indicators was impossible, the results on well-studied representatives of those groups were considered enough to estimate the sterilization efficacy of gClO_2_ using the proposed method.

### 2.2. Incubator

The incubator used was a stainless-steel incubator with a transparent lid. The dimensions were 280 mm for the width, 216 mm for the depth, and 306.5 mm for the average height. The volume was 0.0185 m^3^. Inside the incubator, a plate-rocking station was used as a tilting station to activate the sterilization. This incubator tilting station included an analogue feedback servo (S1213, Batan, Hong Kong, China) for the rocking station actuation.

The automated incubator comprised several gas inlets and one gas outlet. Gas inlets and the gas outlet were managed via electro-valves and could be managed individually by the automated incubator control system. Gas inlets were connected to the laboratory nitrogen gas source and filtered pressurized air from the laboratory. Gas inlets and the outlet were connected with 0.22 µm filter units. The incubator outlet was connected to activated charcoal filters. Activated charcoal filters were advised by the ClO_2_ powder tablet supplier (ClorDiSys Solution, Branchburg, NJ, USA) to absorb residual gClO_2_.

### 2.3. Gaseous ClO_2_ Generation and Detection

For the generation of gClO_2_, a commercial microplate format (i.e., SBS format) plastic custom sterilization plate was used to hold the necessary material. The plate comprised a water reservoir, a powder reservoir for the tablet, and a slope between the two reservoirs (Appendix A). The plate was manufactured in ABS using a Fortus 3D printer (Stratasys, Eden Prairie, MN, USA). The plate was loaded with 17 mL of water in the reservoir and the desired fraction of the solid powder tablet (ClorDiSys Solution, Branchburg, NJ, USA) in the solid tablet container. Then, the plate was placed on the rocking station. After closing the incubator door and all inlets, the rocking station could be activated to tilt. To avoid any potential overpressure within the incubator chamber, the gas outlet remained open during the gas generation. Once ready, the plate was tilted automatically through an external electronic control, thus leading the water to the fraction of the solid tablet, generating gClO_2_ in an exogenous (*ΔH* = −83 kJ/mol at 25 °C) and spontaneous (*∆G* = −162 kJ/mol at 25 °C) reaction (1) (Figure 1). The highest concentration measured during dissolution of one whole ClO_2_ tablet (3.84 g) in water for an approximately equivalent volume of the incubator was 1660 PPM (personal communication from ClorDiSys Solution, Branchburg, NJ, USA). The weight of 3.84 g is the sum of all tablet components (1):5NaClO_2_ + 4NaHSO_4_→4ClO_2_ + 4Na_2_SO_4_ + 2H_2_O + NaCl(1)

After the sterilization, the air inside the incubator was renewed to remove residual ClO_2_ and avoid further toxicity for the following tests. The ventilation process consisted of opening the outlet and the nitrogen or pressurized air gas inlet, pushing the gases inside the incubator to the outlet and renewing the air (or gas mixture) inside the incubator. Concentrations of residual ClO_2_ forms inside the incubator were measured after the ventilation using a calibrated ClO_2_ detector (X-am 5600, Dräger, Lübeck, Germany) one hour and thirty minutes after the end of the ventilation. During the entire test, the same detector was used to measure for concentrations of ClO_2_ formed outside the incubator in the laboratory. Finally, the liquid left in the powder reservoir of the sterilization plate was disposed of, as per local chemical waste management rules, and the sterilization plate was cleaned with water and reused for subsequent sterilization tests.

### 2.4. Sterilization Experiment

#### 2.4.1. Sterilization Protocol

Before exposure, 200 µL of biological indicator solution was inoculated on 35 mm-diameter Petri dishes with vents. For each biological proxy and each tested concentration, the same source batch was used for the control and assessment groups. The control group was not exposed to gClO_2_. The treated group was exposed to gClO_2_. Petri dishes from the treated group were transferred to the incubator in various positions. For quantitative assessment, for each concentration of gClO_2_ tested and the biological indicator, Petri dishes were installed inside the incubator, two on the top with one open and one with the Petri dish lid closed, and two at the bottom inside the incubator, with one open and one with the Petri dish lid closed. gClO_2_ was generated inside the incubator containing the Petri dishes from the treated group (Figure 2). To ensure gas homogeneity, the incubator fan was switched on at least 1 min before the sterilization activation and left on during the entire duration of the sterilization. The sterilization duration was 15 min. Petri dishes from the control group were left outside the incubator in the biosafety cabinet.

After gClO_2_ exposure, the incubator’s air was ventilated for 15 min with pressurized gas from the laboratory supply line set at 0.4 bar relative to the ambient pressure. Subsequently, exposed Petri dish samples were pooled before serial dilution to minimize manipulations, as preliminary experiments showed no visible colony-forming unit (CFU) difference for plates placed at different locations inside the incubator. The viability and survival of test microorganisms were assessed using culture on the biological proxies’ respective media. Results are expressed as CFU/mL.

#### 2.4.2. Biological Indicator Sterilization

The efficacy against *Bacillus subtilis* was examined using a commercial *B. subtilis* spore (Ref. 1.10649.0001, Merck, Darmstadt, Germany) solution stored in the fridge. The same batch was used for all the gClO_2_ tests. Sterilization tests (Section 2.4.1) were carried out using ClO_2_ powder 0 g (0 PPM), 0.15 g (64 PPM), 0.27 g (116 PPM), 0.59 g (255 PPM), and 1 g (433 PPM). Plating of *B. subtilis* was made on Luria agar (LA). Petri dishes were incubated at 37 °C for 48 h.

*Saccharomyces cerevisiae* cells were cultured at 25 °C on a solid yeast extract–peptone–dextrose (YPD) medium. A colony was picked, resuspended in phosphate-buffered saline (PBS), and aliquoted into Petri dishes. Sterilization tests (Section 2.4.1) were carried out using ClO_2_ powder at 0 g (0 PPM), 0.15 g (64 PPM), 0.27 g (116 PPM), 0.5 g 0.59 g (255 PPM), 1 g (433 PPM), and 1.5 g (649 PPM). Finally, Petri dishes were incubated at 30 °C for 72 h on YPD.

Finally, T7 phages were cultivated with their *E. coli* host alive at room temperature. After lysis of the bacterial culture, the lysate was filtered through 0.22 µm pore size syringe filters. The stock obtained was aliquoted and stored in the fridge at 4 °C. T7 was aliquoted into Petri dishes. Sterilization tests (Section 2.4.1) were carried out using ClO_2_ powder at 0 g (0 PPM), 0.15 g (64 PPM), 0.27 g (116 PPM), and 0.59 g (255 PPM). Finally, colonies were counted after incubation at 37 °C for 48 h in double-layer agar (LA; *E. coli*-inoculated LA top agar + T7 sample).

All statistical analyses were performed with R (version 4.2.2, R Foundation for Statistical Computing, Vienna, Austria) and RStudio (version 2023.09.1, RStudio, Inc., Boston, MA, USA). A normality test was first performed for each sample type using the Shapiro–Wilks test. For normally distributed populations, the Student’s t-test was used to compute the *p*-values between the dataset with 0 PPM ClO_2_ and the other concentrations of ClO_2_. Otherwise, the Wilcoxon test was used to compute the *p*-values between the dataset with 0 PPM ClO_2_ and the other concentrations of ClO_2_.

## 3. Results

### 3.1. gClO_2_ Formation Result

After activation, an examination of the powder reservoir revealed that bubbles were visible directly after activation and during 4 to 20 s, depending on the powder quantity (Figure 3). Bubbles were considered enough to deduce the successful generation of the gClO_2_. Also, the typical yellow color indicated the presence of ClO_2_ ions in the solution.

### 3.2. Sterilization Result

Preliminary experiments with *B. subtilis* spores showed that after 15 min of gClO_2_ exposure, plate and plaque counts exhibited no visible CFU differences for plates placed at the top or bottom of the incubator. Similarly, no differences were visible for opened or closed Petri dishes. This emphasized that the gas circulated well in the incubator and reached all samples, even those in closed Petri dishes placed far from the gClO_2_ gas source. Consequently, survival values were monitored using the pooled samples from the top, bottom, and open and closed Petri dishes (see the Section 2 for details).

For all biological indicators, the CFU counts were inversely proportional to the gClO_2_ concentration. The sterility assurance level (SAL) for sterilization should be 10^−6^ [13], also called six-log reduction. The sterilization process afforded fungi a six-log reduction for 433 PPM over 15 min. For viruses, the sterilization process yielded a six-log reduction (2.45 × 10^8^ CFU/mL for 0 PPM to 101 CFU/mL for 443 PPM gClO_2_) for 433 PPM over 15 min. Therefore, successful six-log reduction sterilization was achieved for all proxies for six levels of serial dilution of 443 PPM and above (Figure 4; Appendix A). This result demonstrated that gClO_2_ could effectively sterilize the entire automated incubator against biological indicators in 15 min and can be generated from a solid powder tablet inside an automated incubator with a rocking station, without needing external devices.

### 3.3. Residual Chlorine Dioxide and Residual Toxicity

The residual chlorine dioxide was measured at 0.05 PPM, which is far lower than the 3 PPM maximum residual concentration accepted after food sterilization [25]. To investigate the potential residual toxicity, as a verification qualitative assessment, *S. cerevisiae* cells were also put into culture directly inside the incubator after 10 min of ventilation post-sterilization and 30 min of waiting time after the ventilation. Compared to the control culture outside the incubator, *S. cerevisiae* grew and formed colonies without noticeable differences (inspection by eye).

## 4. Discussion

We demonstrated that 1 g of powder, or 443 PPM, effectively sterilized the incubator, with a sterility assurance level of 10^−^⁶. We observed that below 0.25 g of powder in 17 mL of water, the concentration of gas generated in the incubator was insufficient to lead to any loss of biological proxy viability. Consequently, the recommended dosage for sterilization is between 433 and 650 PPM for 15 min, and the total powder should be above 0.25 g for 17 mL. This recommended gClO_2_ concentration is within the recommended range for sterilization for other fields of application and other volumes [26].

Residual toxicity for cell culture was only briefly qualitatively examined with *S. cerevisiae*, and epithelial cells isolated from human colon tissue with colorectal adenocarcinoma (CACO2 HTC-37) were expanded (Figure 5). Quick examination by eye and double time until cell confluence did not show any difference compared to culture in the regular non-ClO_2_-sterilized incubator. Also, in the literature, ClO_2_ toxicity was studied in vivo in rats [27]. Still, those data were insufficient to investigate whether the residual toxicity is acceptable for human cell-culture-based tests. Therefore, a quantitative assessment with robust immortalized human cell culture and fragile human cell culture should be performed to confirm that the residual toxicity does not impact human cell-culture-based tests.

## Figures and Tables

**Figure 1 pathogens-13-01024-f001:**
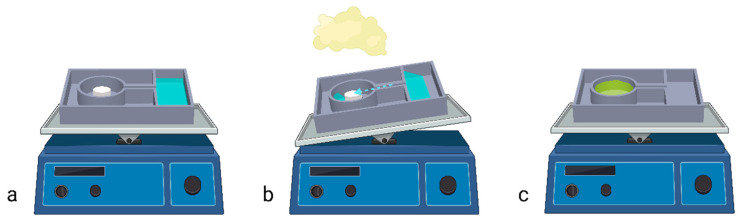
Schematic of the sterilization activation. (**a**) The initial load before activation was visualized (blue: water; yellow: ClO_2_ tablet). (**b**) The rocking station tilts in motion and pours the water on the tablet to generate the chlorine dioxide gas. (**c**) After activation, water with by-products remains.

**Figure 2 pathogens-13-01024-f002:**
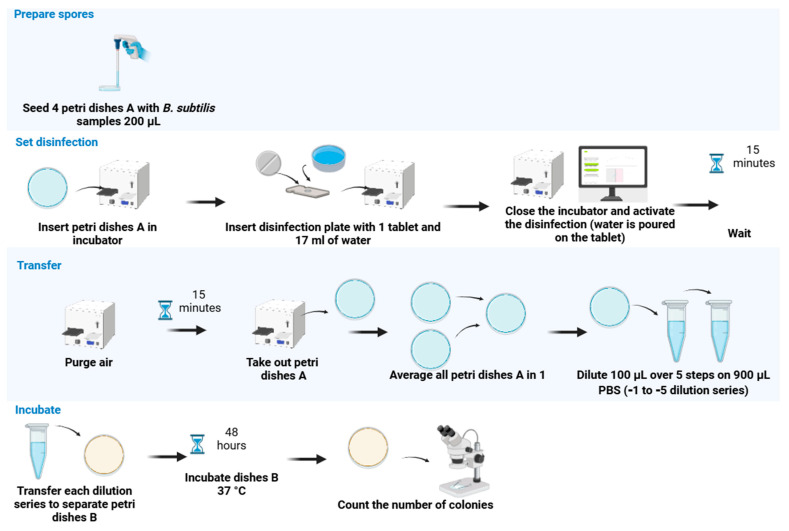
This figure shows the experimental protocol for the sterilization of *B. subtilis*. The incubation time, temperature, and culture medium differed for *S. cerevisiae* (Section 2.4.2. Biological Indicator Sterilization) and T7 (Section 2.4.2. Biological Indicator Sterilization). The hourglass represents the passing of time.

**Figure 3 pathogens-13-01024-f003:**
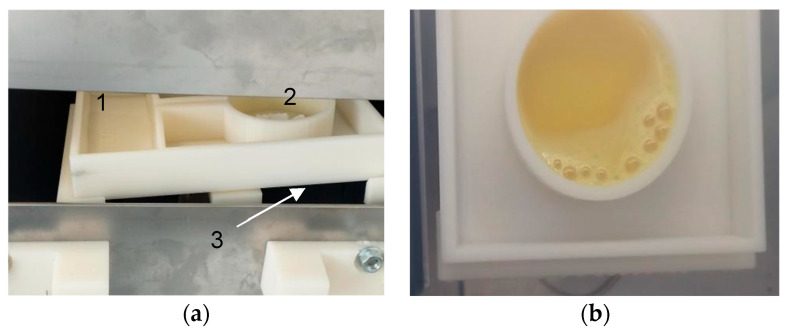
(**a**) Before activation, the sterilization plate was loaded with water (17 mL) (1) and a fraction of a powder tab (here 2 g) (2) and was placed inside the incubator on the rocking station (3). (**b**) After activation of chlorine dioxide powder with water, gaseous chlorine dioxide (gClO_2_) was produced.

**Figure 4 pathogens-13-01024-f004:**
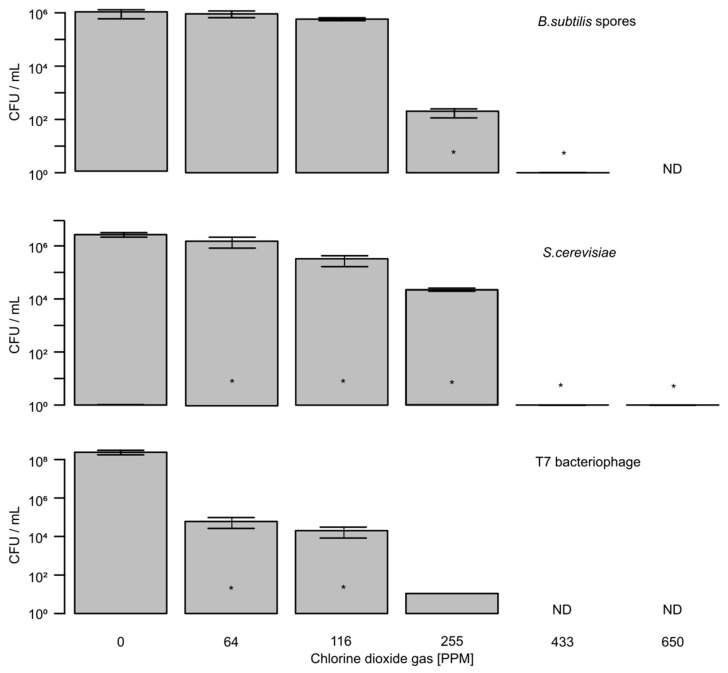
Bacterial spores, fungi, and viruses’ colony-forming units (CFU)/mL with SD (when applicable) for different chlorine dioxide gas (gClO_2_) concentrations. SD: standard deviation. ND: not determined. *: *p*-value < 0.05 between 0 PPM and assessed ClO_2_ concentration.

**Figure 5 pathogens-13-01024-f005:**
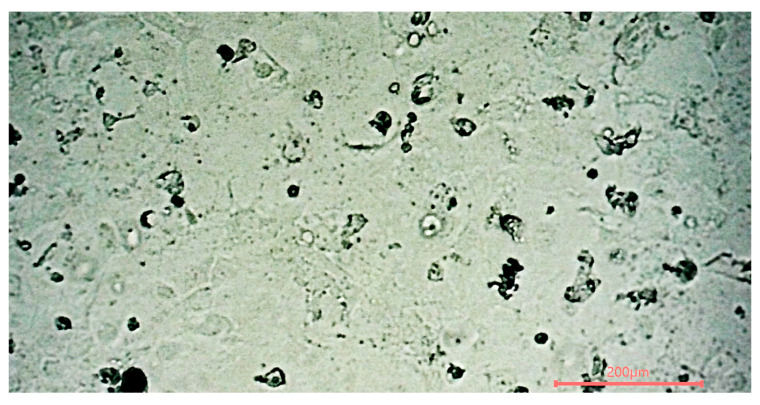
Photo of the human colon cells cultivated post-sterilization under the microscope.

## Data Availability

The original contributions presented in the study are included in the article/Appendix A. Further inquiries can be directed to the corresponding author.

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
