# Peer review of "Method to Generate Chlorine Dioxide Gas In Situ for Sterilization of Automated Incubators"

_pathogens, 2024, doi:10.3390/pathogens13111024_

Round 1

Reviewer 1 Report

Comments and Suggestions for Authors

The topic of the work is very interesting and I am convinced that it will find many recipients. For several decades, chlorine dioxide has been of interest to scientists and is commonly used for disinfection as an aqueous solution and gas, and the presented studies on automatic decontamination with gaseous ClO2 are a very interesting solution that may have practical applications. However, the work lacks a proper introduction supported by literature data. The authors cite only 3 references confirming the effectiveness of eliminating individual microorganisms, while there are many more of these studies, and many of them were published in the last 5 years, and refering to these papers would emphasize the relevance of the reviewed manuscript. In addition, it is worth adding information about the disinfection methods used using ClO2 in the introduction, providing details such as disinfection/sterilization time, ClO2 concentration, its form (solution or gas), and effectiveness against microorganisms. This would allow the reader to better understand the method described.

It is also necessary to enrich the references. 12 items are definitely too few for a scientific paper, and only 4 of them are papers from the last 10 years.

Please check the manuscript carefully - Latin phrases should be written in italics. Similarly with punctuation and units - please check carefully and use the correct forms.

Figure 2 is difficult to read, it is worth considering enlarging it.

The methodology lacks references. They should also be described in more detail.

The discussion lacks references to other studies.

Please correct the references according to the publisher's requirements.

Author Response

Pathogens Rebuttal Reviewer 1

Dear Reviewer,

We appreciate your thorough review and constructive feedback on our manuscript titled "Method to generate chlorine dioxide gas in situ for sterilization of automated incubators." We are grateful for your positive remarks regarding the interest and potential practical applications of our study. We have carefully considered your comments and have made the following revisions to address your concerns:

1)“The topic of the work is very interesting and I am convinced that it will find many recipients. For several decades, chlorine dioxide has been of interest to scientists and is commonly used for disinfection as an aqueous solution and gas, and the presented studies on automatic decontamination with gaseous ClO2 are a very interesting solution that may have practical applications. However, the work lacks a proper introduction supported by literature data. The authors cite only 3 references confirming the effectiveness of eliminating individual microorganisms, while there are many more of these studies, and many of them were published in the last 5 years, and refering to these papers would emphasize the relevance of the reviewed manuscript.

We agree that the literature could have been increased. We have significantly expanded the introduction section to include a more comprehensive review of the literature on chlorine dioxide (ClOâ‚‚) and its applications in disinfection and sterilization. We have added several recent studies to provide a broader context and emphasize the relevance of our work. Specifically, we have included references to studies published in the last five years that demonstrate the effectiveness of ClOâ‚‚ against various microorganisms.

We have enriched the references section, increasing the total number of references to 24, with a focus on including more recent studies. This ensures that our manuscript is well-supported by current literature.

2)” In addition, it is worth adding information about the disinfection methods used using ClO2 in the introduction, providing details such as disinfection/sterilization time, ClO2 concentration, its form (solution or gas), and effectiveness against microorganisms. This would allow the reader to better understand the method described.”

We agree that disinfection/sterilization time from other studies could help readers. We have added disinfection/sterilization time,  and effectiveness against microorganisms. This comparison helps readers better understand the advantages and limitations of each method.

3)” Please check the manuscript carefully - Latin phrases should be written in italics. Similarly with punctuation and units - please check carefully and use the correct forms.”

We agree that Latin phrases should be written in italics. We have carefully reviewed the manuscript to ensure that Latin phrases and variables are written in italics.

4) “Figure 2 is difficult to read, it is worth considering enlarging it.”

We agree that Figure 2 was difficult to read. We have reworked and enlarged Figure 2 to make it easier to read and have adjusted the layout to ensure that titles are on top of the images. This should improve the clarity and readability of the figure.

5)” Please correct the references according to the publisher's requirements.”

We agree that the reference formatting was incorrect. Thank you very much. We have ensured that all references are formatted according to the publisher's requirements.

We believe that these revisions have significantly improved the manuscript and addressed all the concerns raised. We appreciate your valuable feedback and hope that the revised manuscript meets your expectations.

Thank you for your time and consideration.

Sincerely,

Reviewer 2 Report

Comments and Suggestions for Authors

This paper reported a protocol for the sterilization of automated incubators with the

Generation of the chlorine dioxide gas.

The procedures of this protocol were detailed and reasonable. The incubation time, temperature, and culture medium were introduced. However, some content needs to be improved to enhance the quality of this paper.

The content of Figure 4 and Table 1 have similar content; please revise them.

For S. cerevisiae, colony-forming units (CFU)/ml at different concentrations were unreasonable. Why did this happen for the 64 and 116 ppm? Why will the results of ND? 

Comments on the Quality of English Language

Minor editing of English language required.

Author Response

Pathogens Rebuttal Reviewer 2

Dear Reviewer,

We appreciate your detailed review and constructive feedback on our manuscript titled "Method to generate chlorine dioxide gas in situ for sterilization of automated incubators." We have carefully considered your comments and have made the following revisions to address your concerns:

  • “The procedures of this protocol were detailed and reasonable. The incubation time, temperature, and culture medium were introduced. However, some content needs to be improved to enhance the quality of this paper. The content of Figure 4 and Table 1 have similar content; please revise them.”

We acknowledge that Figure 4 and Table 1 contain similar content. To enhance the clarity and conciseness of the manuscript, we have moved Table 1 to the supplementary materials section. This change ensures that the main text is not redundant and allows readers to access detailed data if needed.

  • For S. cerevisiae, colony-forming units (CFU)/ml at different concentrations were unreasonable. Why did this happen for the 64 and 116 ppm? Why will the results of ND?”

We understand your concern regarding the CFU/ml values for S. cerevisiae 1) at different concentrations (255 ppm), and 2) missing concentrations (64 and 116 ppm).

Therefore, we performed the experiment again and 1) removed the previous S. cerevisiae data for 216 ppm and 2) completed the missing data (64 and 116 ppm).

The "ND" (Not determined) results indicate that the experiment was not performed as the necessary reduction for sterilization was already obtained. We have clarified this point in the revised manuscript to ensure that readers understand the variability and limitations of the sterilization process at lower concentrations.

We have reviewed the entire manuscript to ensure that all methodological details are clearly described and that the results are accurately presented.

We believe that these revisions have significantly improved the manuscript and addressed all the concerns raised. We appreciate your valuable feedback and hope that the revised manuscript meets your expectations.

Thank you for your time and consideration.

Sincerely,

Round 2

Reviewer 1 Report

Comments and Suggestions for Authors

The authors have made the suggested corrections, but they are not entirely satisfactory. It is worth expanding the introduction with more than just 3 references. I suggest referring to publications on the effectiveness of ClO2 in relation to various microorganisms and uses. Below are examples of publications on this topic. 

Krüger, T.I.M.; Herzog, S.; Mellmann, A.; Kuczius, T. Impact of Chlorine Dioxide on Pathogenic Waterborne Microorganisms Occurring in Dental Chair Units. Microorganisms 2023, 11, 1123. https://doi.org/10.3390/microorganisms11051123 Yingcai Tang, Yin-Hu Wu,

Hao-Bin Wang, Zhuo Chen, Wen-Long Wang, Xin-Ye Ni, Ao Xu, Hong-Ying Hu, Disinfection-residual bacteria (DRB) after chlorine dioxide treatment: Microbial community structure , regrowth potential, and secretion characteristics, Journal of Hazardous Materials, Volume 476, 2024, 135136, ISSN 0304-3894, https://doi.org/10.1016/j.jhazmat.2024.135136.

Jefri UHNM, Khan A, Lim YC, Lee KS, Liew KB, Kassab YW, Choo CY, Al-Worafi YM, Ming LC, Kalusalingam A. A systematic review on chlorine dioxide as a disinfectant. J MedLife. 2022 Mar;15(3):313-318. doi: 10.25122/jml-2021-0180. PMID: 35449999; PMCID: PMC9015185.

Author Response

Dear Reviewer,

Thank you very much for your valuable feedback and suggestions on our manuscript "Method to generate chlorine dioxide gas in situ for sterilization of automated incubators." We appreciate the time and effort you have taken to review our work and provide constructive comments.

Point 1) “The authors have made the suggested corrections, but they are not entirely satisfactory. It is worth expanding the introduction with more than just 3 references. I suggest referring to publications on the effectiveness of ClO2 in relation to various microorganisms and uses. Below are examples of publications on this topic. Krüger, T.I.M.; Herzog, S.; Mellmann, A.; Kuczius, T. Impact of Chlorine Dioxide on Pathogenic Waterborne Microorganisms Occurring in Dental Chair Units. Microorganisms 2023, 11, 1123. https://doi.org/10.3390/microorganisms11051123 Yingcai Tang, Yin-Hu Wu, Hao-Bin Wang, Zhuo Chen, Wen-Long Wang, Xin-Ye Ni, Ao Xu, Hong-Ying Hu, Disinfection-residual bacteria (DRB) after chlorine dioxide treatment: Microbial community structure , regrowth potential, and secretion characteristics, Journal of Hazardous Materials, Volume 476, 2024, 135136, ISSN 0304-3894, https://doi.org/10.1016/j.jhazmat.2024.135136. Jefri UHNM, Khan A, Lim YC, Lee KS, Liew KB, Kassab YW, Choo CY, Al-Worafi YM, Ming LC, Kalusalingam A. A systematic review on chlorine dioxide as a disinfectant. J MedLife. 2022 Mar;15(3):313-318. doi: 10.25122/jml-2021-0180. PMID: 35449999; PMCID: PMC9015185.

In response to your suggestion to expand the introduction with more references, we have revised the introduction section of our manuscript. We have included additional references that discuss the effectiveness of chlorine dioxide (ClOâ‚‚) in relation to various microorganisms as you suggested. Specifically, we have incorporated the references that you mentioned.

We believe that these additions significantly enhance the introduction and provide a more robust context for our study. We hope that these revisions meet your expectations and improve the overall quality of our manuscript.

Thank you once again for your insightful comments and suggestions. We look forward to your feedback on the revised manuscript.